# Implications and Consequences of SL(2R) as Invariance Group in the Description of Complex Systems Dynamics from a Multifractal Perspective of Motion

**DOI:** 10.3390/e24040484

**Published:** 2022-03-30

**Authors:** Lucian Dobreci, Oana Rusu, Decebal Vasincu, Mihaela Jarcău, Cristina Marcela Rusu, Silviu Gurlui, Vlad Ghizdovat, Alina Gavrilut, Maricel Agop

**Affiliations:** 1Department of Physical and Occupational Therapy, “Vasile Alecsandri” University of Bacau, 600115 Bacau, Romania; luciandd@yahoo.com; 2Faculty of Material Science and Engineering, “Gheorghe Asachi” Technical University, 700050 Iasi, Romania; oana.rusu@academic.tuiasi.ro; 3Faculty of Dental Medicine, “Grigore T. Popa” University of Medicine and Pharmacy of Iași, 700050 Iasi, Romania; decebal.vasincu@umfiasi.ro; 4Faculty of Food Engineering, Stefan cel Mare University of Suceava, 720229 Suceava, Romania; mihaela.jarcau@fia.usv.ro; 5Physics Department, “Gheorghe Asachi” Technical University, 700050 Iasi, Romania; cristina.rusu@tuiasi.ro; 6Physics Faculty, Alexandru Ioan Cuza University of Iași, 700050 Iasi, Romania; sgurlui@uaic.ro; 7Department of Biophysics and Medical Physics, “Grigore T. Popa” University of Medicine and Pharmacy of Iasi, 700115 Iasi, Romania; vlad.ghizdovat@umfiasi.ro; 8Mathematics Faculty, Alexandru Ioan Cuza University of Iași, 700050 Iasi, Romania; gavrilut@uaic.ro; 9Romanian Scientists Academy, 54 Splaiul Independentei, 050094 Bucharest, Romania

**Keywords:** SL(2R) group, invariance group, joint invariant functions, canonical formalism, transitivity manifolds, multifractality

## Abstract

Possible implications and consequences of using SL(2R) as invariance groups in the description at any scale resolution of the dynamics of any complex system are analyzed. From this perspective and based on Jaynes’ remark (any circumstance left unspecified in the description of any complex system dynamics has the concrete expression in the existence of an invariance group), in the present paper one specifies such unspecified circumstances that result directly from the consideration of the canonical formalism induced by the SL(2R) as invariance group. It follows that both the Hamiltonian function and the Guassian distribution acquire the status of invariant group functions, the parameters that define the Hamiltonian acquire statistical significances based on a principle of maximizing informational energy, the class of statistical hypotheses specific to Gaussians of the same average acts as transitivity manifolds of the group (transitivity manifolds which can be correlated with the multifractal-non-multifractal scale transitions), joint invariant functions induced through SL(2R) groups isomorphism (the SL(2R) variables group, and the SL(2R) parameters group, etc.). For an ensemble of oscillators of the same frequency, the unspecified circumstances return to the ignorance of the amplitude and phase of each of the oscillators, which forces the recourse to a statistical ensemble traversed by the transformations of the Barbilian-type group. Finally, the model is validated based on numerical simulations and experimental results that refer to transient phenomena in ablation plasmas. The novelty of our model resides in the fact that fractalization through stochasticization is imposed through group invariance, situation in which the group’s transitivity manifolds can be correlated with the scale resolution.

## 1. Introduction

The measurable Lie groups theory is a fundamental tool in integral geometry, more precisely in the production of the so-called geometric probabilities [1,2]. Although the concept of geometric probability has a “long history”, only recently it has been realized that through the measurable groups theory this concept can be applied for solving physical problems, such as some experimental problems displaying statistical interference [3,4]. This is due to the fact that, according with Jaynes’s observations [5,6], any unspecified circumstance in an experiment has the specific expression in the existence of an invariance group. This means that the respective circumstance is not left unspecified because of subjective reasons, but because in the experiment the circumstance does not manifest. For example, in a Coulomb-type experiment, the force must be an orthogonal invariant, since, no matter the direction we place the electrical charges with respect to each other, at equal distance, the force value is the same. Therefore, the probabilities determined through global invariance can, and must, display a frequential meaning. They are numerically equal with the induced frequencies, when we operate with the least amount of constraints, i.e., in the most natural framework of a reproducible phenomenon. We are thus being led to a direct correspondence with statistical physics [7].

From such a perspective, various operational procedures such as group invariants, joint invariant functions, Riemannian type differential geometries, parallel transport of directions in the Levi-Civita sense, harmonic mappings from the usual space to the hyperbolic one become functional in the description of the dynamics of various complex systems. For instance, in the recent papers [8,9,10], assimilating any complex system with a multifractal mathematical object, non-differentiable behaviors of complex systems are analyzed. Then in-phase coherence of the dynamics of any complex system structural units, which implies, for example, special cubics with SL(2R)-type group invariance, special differential geometry of Riemann type associated to such cubics, special apolar transport of cubics assimilated with parallel transport of direction in Levi-Civita sense, special harmonic mapping imposed by means of parallel transport of direction, etc., make possible various scenarios to chaos (period doubling, intermittences, etc.), without going into chaos.

In the present paper, we analyze implications and consequences of using SL(2R) as invariance groups in the description (at any scale resolution) of the dynamics of any complex system. Such an approach was possible since various operational procedures (group invariances, Riemann type differential geometries, harmonic mappings based on variational principles, etc.) are induced by the motion laws which characterize the complex systems dynamics (invariant laws with respect to spatial and time co-ordinates transformations, but also to scale transformations).

## 2. A Short Reminder on Measurable Lie Groups as Invariance Groups

A Lie group is measurable if it admits a singular integral invariance function up to a multiplicative constant [11,12]. The integral of that function on the group’s variables domain is the group’s measure. We will thus try to determine how can the integral invariant function of a group be defined, in order to provide the possibility of constructing measures having the role of probabilities.

By definition, being given a Lie group with r parameters in n variables, through relations
(1)$$y^{i}=f^{i}\left(x^{1},\ldots ,x^{n},a^{1},\ldots ,a^{r}\right)\;i=1,\ldots ,n$$
it can be said that the function $F\left(x^{1},\ldots ,x^{n}\right)$ is an integral invariant function on the group if
(2)$$\int_{D_{x}}F\left(x^{1},\ldots ,x^{n}\right)dx^{1}\ldots dx^{n}=\int_{D_{t}}F\left(y^{1},\ldots ,y^{n}\right)dy^{1}\ldots dy^{n}$$
for every domain $D_{x}$, for which the integral makes sense. By explicating Equation (1) around the transformation identity that we admitted for parameters variables $a^{1}=a^{2}=\ldots =a^{r}=0$, we can find the conditions that must be satisfied by F for it to be an integral invariant function.

Indeed, the transformations neighboring the unit will be [11,12]:(3)$$y^{i}=x^{i}+\xi_{h}^{i}\left(x\right)a^{h}$$
where we make use of the repeatable indices addition. In this case, the transformation Jacobian is
(4)$$J=1+\frac{\partial \xi_{h}^{i}\left(x\right)}{\partial x^{i}}a^{h}$$

In the same approximation, we can develop the function $F\left(y^{1},\ldots ,y^{n}\right)$ around the parameters’ origin, and we will get
(5)$$F\left(y^{1},\ldots ,y^{n}\right)=F\left(x^{1},\ldots ,x^{n}\right)+\frac{\partial F}{\partial x_{i}}\xi_{h}^{i}\left(x\right)a^{h}$$

On the other hand, the defining condition (2) of the integral invariant function comes back to
(6)$$F\left(y^{1},\ldots ,y^{n}\right)=JF\left(x^{1},\ldots ,x^{n}\right)$$
and taking into account Equations (4) and (5), it becomes:(7)$$\frac{\partial }{\partial x^{i}}\left[\xi_{h}^{i}\left(x\right)F\left(x\right)\right]$$
where we neglected the second order terms in $a^{k}$ parameters.

Therefore, for a function $F\left(x\right)$ to be an integral invariant function of group (1), it must be a solution of the partial derivatives equations system (7), named Deltheil’s equations system [13]. The result is known as Deltheil Theorem and it is of maximum importance in geometric probabilities theory [14]. Through Deltheil’s equations the integral invariant function is, in principle, known as soon as the infinitesimal transformations (3) of the group in question are known.

## 3. SL(2R) as an Invariance Group

We will be practically interested in only one group, namely the so called centro-affine unimodular group, i.e., the SL(2R) [15]. This is the homogenous transformations of the affine plane group, which in various areas, represented by real unimodular matrices
(8)$$\begin{array}{l} x^{\prime }=\alpha x+\beta y\\ y^{\prime }=\gamma x+\delta y\\ \alpha \delta -\beta \gamma =1 \end{array}$$

This is a group in two variables with three parameters, because only three from α, β, γ, and δ are independent, by virtue of the fact that the matrix determinant is the unit. We adopt the following parametrization for the infinitesimal transformations corresponding to (3)
(9)$$\alpha =1+\frac{1}{2}a^{2};\;\beta =a^{1};\;\gamma =-a^{3};\;\delta =1-\frac{1}{2}a^{2}$$
case in which these become:(10)$$\begin{array}{l} x^{\prime }=x+ya^{1}+\frac{x}{2}a^{2}\\ y^{\prime }=y-\frac{y}{2}a^{2}-xa^{3} \end{array}$$

From these relations we can observe that the group’s action is transitive: there is at least one transformation of the group that links two points of co-ordinates $\left(x,y\right)$ and $\left(x^{\prime },y^{\prime }\right)$, respectively. This can be observed from the fact that Equation (10) in unknowns $a^{1},\;a^{2},\;\mathrm{and}\;a^{3}$ form a compatible system. Actually, there is a simple infinity of such transformations, which means that the action of this group is multiple transitive.

Now we can directly read from Formulas (10) the matrix ξ which characterizes the infinitesimal transformations, necessary in Deltheil’s equation
(11)$$\begin{array}{l} \xi_{1}^{1}=y;\;\xi_{2}^{1}=\frac{x}{2};\;\xi_{3}^{1}=0\\ \xi_{1}^{2}=0;\;\xi_{2}^{2}=-\frac{y}{2};\;\xi_{3}^{2}=-x \end{array}$$

Prior to writing Deltheil’s equations, let us observe that this picture gives us the basis of the Lie algebra associated to the SL(2R) group, given through the infinitesimal generators
(12)$$X_{h}=\xi_{h}^{i}\frac{\partial }{\partial x^{i}}$$
i.e.,
(13)$$X_{1}=y\frac{\partial }{\partial x};\;X_{2}=\frac{1}{2}\left(x\frac{\partial }{\partial x}-y\frac{\partial }{\partial y}\right);\;X_{3}=-x\frac{\partial }{\partial y}$$

These infinitesimal operators give the following group structure, through the associated Poisson parentheses (or the operators commutators)
(14)$$\left[X_{1},X_{2}\right]=X_{1};\;\left[X_{2},X_{3}\right]=X_{3};\;\left[X_{3},X_{1}\right]=-2X_{2}$$
where we noted with square parentheses the commutators of the respective operators. We can thus find the structure constants of the SL(2R) group algebra
(15)$$C_{12}^{1}=C_{23}^{3}=1;\;C_{31}^{2}=-2$$
all the other ones being null.

Taking into account (11), the Deltheil’s equations characteristic to the group are
(16)$$\begin{array}{l} y\frac{\partial F}{\partial x}=0\\ \frac{\partial }{\partial x}\left(xF\right)-\frac{\partial }{\partial y}\left(yF\right)=0\\ x\frac{\partial F}{\partial F}=0 \end{array}$$
and adopt the solution F=const., i.e., we can take F=1. Therefore, the SL(2R) group is measurable, having the elementary measure, which we previously discussed,
(17)$$d\mu \left(x,y\right)=dx\Lambda dy$$
where with Λ we noted the exterior product of the differential form. According to Jaynes’ observations [5], if unspecified circumstances which admit this invariance group exist, then the probable a priori equal situations accept a uniform distribution of an elementary measure given by (17). We wish now to show such an unspecified circumstance, directly resulting from taking into account the canonic formalism.

Indeed, the fact that SL(2R) invariates the 2-form (17) shows that it is, in equal measure, a simplectic group. The corresponding Hamiltonian dynamics is generated in the “tangent” space through the vectors $X_{1},\;X_{2},\;\mathrm{and}\;X_{3}$ from (13), which satisfies the commutation relations (14). A general “tangent” vector is a linear combination of the form
(18)$$T=aX_{1}+2bX_{2}+cX_{3}$$
and poses the problem of finding the invariant functions along the trajectories tangent to this vector, i.e., the solutions of equation
(19)$$TH\left(x,y\right)=0$$

Taking into account (13), this equation can be explicitly written
(20)$$\left(bx+ay\right)\frac{\partial H}{\partial x}-\left(cx+by\right)\frac{\partial H}{\partial y}=0$$

The differential system characteristic to this equation is
(21)$$\frac{dx}{bx+ay}=-\frac{dy}{cx+by}=\mathrm{dt}$$
and admits the immediate prime integral
(22)$$H\left(x,y\right)=\frac{1}{2}\left(cx^{2}+2bxy+ay^{2}\right)$$

It follows that the solution of Equation (20) will be a random function of this formation which plays a special role in the theory described here, namely, that of a Hamiltonian considered as a motion generator. Indeed, the differential systems (21) is the Hamilton equations systems associated to (22), i.e.,
(23)$$\frac{dx}{dt}=\frac{\partial H}{\partial y};\;\frac{dy}{dt}=-\frac{\partial H}{\partial x}$$
in the case in which x is a co-ordinate and y is the impulse of the considered system, for example a harmonic oscillator. Here we noted with dt the common value of the two differentials from (21), i.e., the differential of the parameter on the vector’s (18) integral curves.

In principle, the Gaussian distribution density can be found among the solutions of Equation (19), expressed as
(24)$$\Gamma \left(x,y\right)=\exp\left\{-H\right\}$$
in which case statistical meanings can be conferred to parameters a,b, and c. This mathematically states the idea of statistical mechanics, according to which the probability density must be a motion integral, but only for the considered particular case, that also includes the important problem of a harmonic oscillator (for other details see [16]).

This, however, does not in any way favor neither the Gaussian probability density, nor the exponential one, because they are decided from other considerations, such as the maximum informational entropy. From this point of view, an observation is absolutely necessary, which directs us, on the one hand, towards another important statistical quantity, the informational energy, and provides us, on the other hand, a certain construction principle for integral invariant functions (for other details see [3,4,10,16]).

If the problem described by Equation (19) is determined from a dynamical point of view, in the sense that the Hamiltonian (22) characterizes a well-established system, the a,b, and c parameters are fixed, and Equation (19) selects a certain class of SL(2R) group trajectories, specified by the afore mentioned parameters values. It is quite easy to provide the trajectories class in question, with the help of system (21). In the hypothesis that a and c are positive, and the quadratic form (22) is positively defined, the system (21) gives the solution
(25)$$\begin{array}{l} x^{\prime }=\left(\cos wt+\frac{b}{a}\sin wt\right)x+\frac{a}{w}\left(\sin wt\right)y\\ y^{\prime }=-\frac{c}{w}\left(\sin wt\right)x+\left(\cos wt-\frac{b}{w}\sin wt\right)y \end{array}$$
where $w=\sqrt{ac-b^{2}}$. In another working hypothesis related to the quadratic form (22) we can, obviously, obtain another evolution matrix, but still unimodular.

Things get quite complicated, however, if the quadratic form (22) is obtained through statistical interference, according to the principle of maximum informational entropy [16], due to the fact that we have here two types of hypotheses: one for averages, the other one for statistical variances, and both are intransitive from a mathematical point of view (for other details see [16]). This means that the variety of hypotheses is not linear, as it happens, for example, in the case where only the averages are specified. There is, however, an important special case, namely, the one in which the Gaussians refer to the same average. In this case, for the inference on ensembles referring to impulse and position, x and y represent the differences of the current quantities towards the average
(26)$$x=q-q_{0}\;\mathrm{and}\;y=p-p_{0}$$

Any statistical hypothesis is specified here through a particular choice of a,b, and c coefficients from the quadratic form (22). The class of all these statistical hypotheses is marked by the invariance of this quadratic form, because the Gaussian distribution density (un-normalized) can be also written as
(27)$$\Gamma^{\prime }\left(x^{\prime },y^{\prime }\right)=\exp\left[-H^{\prime }\left(x^{\prime },y^{\prime }\right)\right]$$
where
(28)$$H^{\prime }\left(x^{\prime },y^{\prime }\right)=\frac{1}{2}\left(a^{\prime }x^{\prime 2}+2bx^{\prime }y^{\prime }+c^{\prime }y^{\prime 2}\right)$$
under the condition
(29)$$H^{\prime }\left(x^{\prime },y^{\prime }\right)=H\left(x,y\right)$$

If $x^{\prime },y^{\prime }$ are linked to x,y through an unimodular transformation
(30)$$\begin{array}{l} x^{\prime }=\alpha x+\beta y\\ y^{\prime }=\gamma x+\delta y \end{array}$$
evidently acceptable because the inference takes place at constant averages, then condition (29) imposes for a,b, and c the following group with three parameters
(31)$$\begin{array}{l} a^{\prime }=\delta^{2}a-2y\delta b+\gamma^{2}c\\ b^{\prime }=-\beta \delta a-\left(\beta \gamma +\alpha \delta \right)b-\alpha \gamma c\\ c^{\prime }=\beta^{2}a-2\alpha \beta b+\alpha^{2}c \end{array}$$

This group is evidently isomorphic with the group (30), yet it has the remarkable property that its action is intransitive on the a,b, and c variables space. In order to practically show what this means, we will adopt for group (30) the parametrization (9) case in which the infinitesimal transformations corresponding to (30) are
(32)$$\begin{array}{l} a^{\prime }=a-aa^{2}+2ba^{3}\\ b^{\prime }=b-aa^{1}+ca^{3}\\ c^{\prime }=c-2ba^{1}+ca^{2} \end{array}$$

Considered as a linear system in $a^{1},\;a^{2},\;\mathrm{and}\;a^{3}$ unknowns, this system is not compatible. Therefore, no transformation can be found that can link the a,b, and c and $a^{\prime },b^{\prime },\;\mathrm{and}\;c^{\prime }$ triplets, which means that the action of the group in the space of these parameters is intransitive, as we said before. This shows that the group effectively acts only under the condition of a certain connection between a,b, and c parameters, that must remain invariant, i.e., on the so-called transitivity manifolds of the group. We can also discover invariant functions by observing that a necessary and sufficient condition for the invariance through group of the function is that it satisfies the partial derivatives equations system
(33)$$\xi_{h}^{i}\frac{\partial F}{\partial x^{i}}=0$$

This can be easily deducted from (5) under the invariance condition $F\left(x\right)=F\left(y\right)$, in the hypothesis that the values of the $a^{h}$ parameters are arbitrary. In our case, it results from (32) that the matrix ξ is given by
(34)$$\begin{array}{l} \xi_{1}^{1}=0;\;\xi_{2}^{1}=-a;\;\xi_{3}^{1}=2b\\ \xi_{1}^{2}=-a;\;\xi_{2}^{2}=0;\;\xi_{3}^{3}=c\\ \xi_{1}^{3}=-2b;\;\xi_{2}^{3}=c;\;\xi_{3}^{3}=0 \end{array}$$
and the system (33) becomes
(35)$$\begin{array}{l} -a\frac{\partial F}{\partial b}-2b\frac{\partial F}{\partial c}=0\\ -a\frac{\partial F}{\partial a}+c\frac{\partial F}{\partial c}=0\\ 2b\frac{\partial F}{\partial b}+c\frac{\partial F}{\partial b}=0 \end{array}$$

The solution of this system is the arbitrary function
(36)$$\Delta =ac-b^{2}$$
so that the transitivity manifolds of group (31) are given by
(37)$$ac-b^{2}=\mathrm{const}.$$

Therefore, the hypotheses class specific to the same average Gaussians is characterized by the property that the discriminant of the quadratic forms (22) is a constant.

For the moment, let us observe another important connection between the theory of measurable continuous groups and the Gaussian construction through statistical inference according to the principle of maximum informational entropy. Firstly, let us notice that we can write, with the help of (34), the infinitesimal generators of the group (31) action. They are
(38)$$\begin{array}{l} A_{1}=-a\frac{\partial }{\partial b}-2b\frac{\partial }{\partial a}\\ A_{2}=-a\frac{\partial }{\partial a}+c\frac{\partial }{\partial c}\\ A_{3}=2b\frac{\partial }{\partial a}+c\frac{\partial }{\partial b} \end{array}$$

These operators satisfy the commutation relations (14), which, obviously, is a normal fact: the two algebras are accomplishments of the one and the same algebra, namely, the SL(2R) group, with structure constants given by (15). The later ones dictate the properties of the respective Lie algebra, the different forms of it in one, two, or three dimensions being only accidental.

It is certain that, considering (37), the group effectively acts in two variables, in other words a,b, and c depend on two parameters. Now we will show the important connection previously deduced: Gaussians can be considered as families of invariant manifolds with three parameters, for the SL(2R) group, having associated the group given by (38), as a parameter group.

## 4. SL(2R) Joint Invariant Functions—On the Parametrization of a Harmonic Oscillators Ensemble

Further details on parameters families of manifolds invariant to groups action can be found in “Integral Geometry” by Stoka [17]. For the current necessities, we will extract only the part that is of importance to us, and that is the fact that group (31) has been produced by induction as a a,b, and c parameters group induced by group (30), under the condition of $H\left(x,y\right)$ functions invariance, given by (29). Reciprocally, taking into account groups (30) and (31) for variables a,b, and c and parameters x,y, respectively, the resulting three parameters invariant manifolds families are, indeed, only the functions $H\left(x,y\right)$. This fact needs to be detailed, because it can be generalized with important physical applications, providing, in particular, some physical meanings to the a,b, and c parameters. The general theory of parametric invariant manifolds families gives this manifold the following solutions for Stoka’s equations [17]
(39)$$X_{i}f\left(x,a\right)+A_{i}f\left(x,a\right)=0$$

Here $X_{i}$ are the infinitesimal generators of the variables group, $A_{i}$ are the ones of the parameters group and it is necessary for the two groups to be isomorphic, i.e., their Lie algebras to belong to the same and only algebra, as was, in our case, (13) and (38). In this case, if we take into account (19) and (38) for $X_{i}$ and $A_{i}$, thus detailing Stoka’s Equation (39), we indeed obtain for the three parameters invariant manifolds families the previous $H\left(x,y\right)$ function, but, and we must emphasize this, only under the transitivity condition (37).

For now, let us keep in mind a very important fact, that Stoka’s equation provide the possibility of constructing a priori invariant measures, through certain groups, depending on certain parameters, under the condition that the variables and parameters groups be isomorphic. Generally speaking, if we have two isomorphic groups, we can find the common invariant manifolds families, without a priori cataloging one group or the other as a parameter group.

In this sense we give now an example of a certain parametrization of an ensemble of harmonic oscillators, starting from the motion equation. In the one-dimensional case, this is the usual equation of the harmonic oscillator
(40)$$\ddot{q}+w^{2}q=0$$
where q is the relevant coordinate. We will write the general solution for this equation as
(41)$$q\left(t\right)=ze^{i\left(wt+\Phi \right)}+\bar{z}e^{-i\left(wt+\Phi \right)}$$
where z is a complex amplitude, $\bar{z}$ is its complex conjugate, Φ is a specific phase, and t is a time parameter. In this way, z and Φ label each oscillator from an interval, eventually an ensemble, which has as a general characteristic the motion Equation (40) and, thus, the same frequency w. This is, for example, the case for a “Planck resonators” ensemble, which interacts with electromagnetic radiation, its analysis leading to the well-known law of energy density distribution by frequencies (for other details see [16]). This ensemble can be described by a continuous group with three variables of three parameters, as follows.

Let us note that the ratio of the fundamental solution of Equation (40), noted with *k*
(42)$$k=e^{2i\left(wt+\Phi \right)}$$
is a solution of the Schwartz equation [16]
(43)$$\left\{k,t\right\}=2w^{2}$$
where
(44)$$\left\{k,t\right\}={\left(\frac{k^{\prime \prime }}{k^{\prime }}\right)}^{\prime }-\frac{1}{2}{\left(\frac{k^{\prime \prime }}{k^{\prime }}\right)}^{2}$$
is the Schwartzian of the k function with respect to the t variable, the accent representing, as usual, the derivation with respect to the variable. Now Equation (43) has the important property that it is invariant with respect to the homographic function transformation, in the sense that the function
(45)$$\sigma \left(t\right)=\frac{\alpha k+\beta }{\gamma k+\delta }$$
satisfies the same equations
(46)$$\left\{\sigma ,t\right\}=2w^{2}$$

Transformations (45) form a three real parameters continuous group on the complex line, which confers to parameter k a projective trait. This allows the introduction of a projective parameter, specific to each oscillator, through relation
(47)$$\sigma \left(t\right)=\frac{z+\bar{z}k}{1+k}$$
which, obviously, satisfies Equation (46). Therefore, between the parameters of various oscillators from the ensemble, a real homographic relation must exist
(48)$$\sigma_{1}\left(t\right)=\frac{\alpha \sigma \left(t\right)+\beta }{\gamma \sigma \left(t\right)+\delta }$$

Group (48) can be considered a sort of “synchronizing” group between various oscillators, process to each, obviously, their amplitudes take part as well, in the sense that they are also correlated, as are their phases. The usual synchronizing through phase delay must be here only a very particular case. Indeed, group (48) implies for z, $\bar{z}$, and k the following parametric group
$$z↔\frac{\alpha z+\beta }{\gamma z+\delta }$$
(49)$$k↔\frac{\gamma \bar{z}+\delta }{\gamma z+\delta }k$$
as it can be easily verified. This shows that, indeed, the phase of k is only shifted by a quantity which depends on the oscillators’ amplitude, and not only that, but the fact that the oscillator’s amplitude is also homographic affected.

Adopting now for group (49) parametrization (9), we obtain the following infinitesimal generators of group (49)
(50)$$\begin{array}{l} B_{1}=\frac{\partial }{\partial z}+\frac{\partial }{\partial \bar{z}}\\ B_{2}=z\frac{\partial }{\partial z}+\bar{z}\frac{\partial }{\partial \bar{z}}\\ B_{2}=z^{2}\frac{\partial }{\partial z}+{\bar{z}}^{2}\frac{\partial }{\partial \bar{z}}+\left(z-\bar{z}\right)k\frac{\partial }{\partial k} \end{array}$$
with commuting relations
(51)$$\left[B_{1},B_{2}\right]=B_{1};\;\left[B_{2},B_{3}\right]=B_{3};\;\left[B_{3},B_{1}\right]=-2B_{2}$$
which show the same structure as the SL(2R) Lie algebra. Therefore, the Lie algebra of group (49) is again a form of the Lie algebra of group SL(2R). Actually, as it can be easily seen, group (49) represents only another action of group SL(2R), made in three variables $z,\bar{z},\;\mathrm{and}\;k$.

These being said, we are now in the conditions of Stoka’s theorem and we can try to find the function which are simultaneously invariant to the action of groups (13) and (50), as solutions for the Stoka’s equations
(52)$$X_{i}f\left(x,y,z,\bar{z},k\right)+B_{i}f\left(x,y,z,\bar{z},k\right)=0,i=1,2,3$$

By explicating these equations with the help of (13) and (50), we can easily solve them by successive reduction, in order to obtain simultaneously invariant functions (joint invariant functions) in the form
(53)$$f\left(\mu ,\nu \right)=\mathrm{const}.$$
where μ and ν have the expressions
(54)$$\begin{array}{l} \mu =\frac{-i\left(z-\bar{z}\right)}{\left(x-zy\right)\left(x-\bar{z}y\right)}\\ \nu =k\frac{x-\bar{z}y}{x-zy} \end{array}$$
from which ν is unimodular complex, and μ is real. A particular class of such invariant functions are the linear combinations of the type
(55)$$p\mu =m\left(\nu +\frac{1}{\nu }\right)+2n$$
where m,n, and p are three real arbitrary constants.

Taking into account (54), Equation (55) transforms into
(56)$$mk^{-1}Z^{2}+2nZ\bar{Z}+mk{\bar{Z}}^{2}\equiv p$$
where
(57)$$Z=\frac{x-\bar{z}y}{\sqrt{-i\left(z-\bar{z}\right)}}$$
admitting that
(58)$$-i\left(z-\bar{z}\right)>0$$

Equation (56) represents a manifold of conics from the $\left(x,y\right)$ plane. These are ellipses if
(59)$$m^{2}-n^{2}<0$$
condition which is always satisfied if
(60)$$\begin{array}{l} m=Q_{1}sh2r\\ n=Q_{1}ch2r \end{array}$$
where $Q_{1}$ is a real constant. We need to consider here a very important particular case, namely the one in which z is purely imaginary, having the value z=i, without narrowing the generality. In this case, the quadratic form (56) can be compared to $H\left(x,y\right)$ from (22), which gives for a,b, and c the following values
(61)$$\begin{array}{l} a=Q_{1}\left(ch2r+sh2r\cos\Phi \right)\\ b=-Q_{1}sh2r\sin\Phi \\ c=Q_{1}\left(ch2r-sh2r\cos\Phi \right) \end{array}$$
where Φ is the value of k, assumed as being fixed. These relations show that, indeed, a,b, and c can be found on trajectories of group (31), because they are finite transformations generated by the infinitesimal transformations (38) of this group, starting from the standard quadratic form of coefficients $a=Q,\;b=0,\;\mathrm{and}\;c=Q_{1}$. Therefore, the square of $Q_{1}$ is precisely the value of constant Δ from (36) which characterizes the transitivity manifolds of group (31). Let us note that, in the case of multifractal dynamics, $Q_{1}$ corresponds to the constant associated to the multifractal-non-multifractal scale transition. In the case of monofractal dynamics described through Peano curves at Compton scale resolution, $Q_{1}$ must be put into correspondence with the Planck constant *h* [16,18]. In Appendix A some correlations between the $Q_{1}$ constant and the scale resolution in various fractal/multifractal dynamics can be found.

It is important to notice that the Gaussian obtained in this way is only a particular case of a distribution which can be extracted, with the condition that it needs to additionally satisfy the principle of maximum informational entropy under quadratic constraints. The solutions of Stoka’s equation can be much more general and can be selected according to other criteria from group theory.

As for the group (49), we must firstly note an historical aspect. This group was discovered by Barbilian as a covariance group for binary cubic forms; later on, it was observed that the group was measurable [6]. By explicating the quantities from (50), Deltheil’s equations become
(62)$$\begin{array}{l} \frac{\partial F}{\partial z}+\frac{\partial F}{\partial \bar{z}}=0\\ z\frac{\partial F}{\partial z}+\bar{z}\frac{\partial F}{\partial \bar{z}}=0\\ z^{2}\frac{\partial F}{\partial z}+{\bar{z}}^{2}\frac{\partial F}{\partial \bar{z}}+\left(z-\bar{z}\right)k\frac{\partial F}{\partial k}=\left(3z-\bar{z}\right)F \end{array}$$
and their solution will be given, up to a multiplicative constant, by
(63)$$F\left(z,\bar{z},k\right)=\frac{1}{{\left(z-\bar{z}\right)}^{2}k}$$

Therefore, the elementary measure of this group will be
(64)$$d\mu \left(z,\bar{z},k\right)=\frac{dz\Lambda d\bar{z}\Lambda dk}{{\left(z-\bar{z}\right)}^{2}k}$$

This group provides meaning to some statistical quantities, through comparison with an oscillators ensemble, each of them being described by three physical quantities (for details see also [16]).

Therefore, Jaynes’ unspecified circumstances for an oscillators ensemble described in Equation (40) come back to not knowing the amplitude and phase for each of them, making it necessary to appeal to the statistical ensemble by explicating its element (the oscillator). This involves quantities $z,\bar{z},\;\mathrm{and}\;k$, the ensemble being traversed with the help of Barbilian’s group transformations. By virtue of this condition, we can say that the elementary probability on this ensemble is given by (64). This means that, if we note with
(65)$$\begin{array}{l} z=u+i\nu \\ k=e^{i\Phi } \end{array}$$
the measure (64) transforms into
(66)$$d\mu =\frac{du\Lambda d\upsilon \Lambda d\Phi }{\nu^{2}}$$

We can see from (66) that the oscillators’ phase is uniformly distributed on the ensemble, just like the real part of the amplitude. As for the imaginary part of the amplitude, it can be said that its inverse is uniformly distributed.

In general, the Barbilian group is not compact, just like the SL(2R) group. This means that the integral of (66) along the entire domain of μ, ν, and Φ is not finite. Therefore, distribution (66) cannot be normalized, with the exception of finite intervals μ and ν, and only in this way it could be used in statistical calculus. This does not in any way diminish the importance of this group, because it can provide a physical meaning, in relation with a harmonic oscillator, to quantities a,b, and c obtained, in general, through statistical inference with respect to the maximum informational entropy.

We have here another example, which illustrates a general fact: the need to realize our degree of ignorance in a certain problem imposes explicating the ensemble described by a usual distribution through the maximum informational entropy. This can be done here by explicating its element in a way which differs from the one previously mentioned, i.e., through the Hamiltonian. Here we were taking into account a oscillator at a given moment in time, in the other case we were discussing about the entire trajectory of a oscillator, for which the Hamiltonian is invariant.

Anyway, an important conclusion can be made: if the principle of maximum informational entropy admits a maximum ignorance in statistical inference, then the parametric continuous groups show in a concrete way what this maximum means; they show what we should know, obviously in the current state of knowledge, they explain the reason for not knowing, and through integral invariant functions, they show how much we can retain from the knowledge process (for other details see also [16]).

## 5. Nonlinear Behaviors in Ablation Plasma Dynamics—Numerical Simulations and Applications

The above presented result specify that, in order to have information about an oscillators ensemble we need to know the amplitude and phase of each oscillator, according to the usual theory of the harmonic oscillator. In this process, it is irrelevant which individual oscillator we choose, because once we know one of them, we can have an a priori knowledge about all of them: the Barbilian transitive group warrants this fact. This however imposes for the a priori probability the previously mentioned invariance.

In such perspective, in the following, let us consider that the previously mentioned oscillators ensemble (self-structured both structural and functional as a multifractal mathematical object) can be identified with an ablation plasma. Then, in accordance with the operation procedures on multifractal manifolds described in [16] (e.g., Riemannian differential geometries, parallel transport of direction in the Levi-Civita sense, harmonic mappings–for other details see Appendix B), the explicit form of parameter *z* from the previous section is given by the relation
(67)z=coshχ2−sinhχ2e−iαcoshχ2+sinhχ2e−iα, α∈ℝ
with α real and arbitrary, as long as χ2 is the solution of a Laplace-type equation for the free space, such that ∇2χ2=0. For a choice of the form α=2ωt, in which case a temporal dependency was introduced in the complex system dynamics, (67) becomes:(68)z=ie2χsin2ωt−sin2ωt−2ieχe2χcos2ωt+1−cos2ωt+1

Numerical simulations were performed for this extension of the multifractal representation of the model. The simulations were performed in Maple software followed by data treatment with OriginPro. The meanings of the employed quantities are the usual ones from [8,9]. The results are presented in Figure 1, where we have represented Re (*F*), Im (*F*) and *F* for a fixed resolution scale dependent parameter (ω) and the fractal time. The use of fractal time becomes mandatory as when investigating transient phenomena as the energy and temporal operating spectra are interconnected. We can observe, for various chosen scales, the shape and frequency of this periodic evolution. The *F* function is defined by a base of period doubling and transitions towards chaotic and modulated dynamics. This incremental change in the periodicity of the system is related to a scattering type process of the oscillators ensemble, where unique kinetic groups are generated and defined during the evolution of the system.

To better understand this effect we have represented the 2D maps for *F(ω,t)* for different maximum values of *ω*. It can be seen that in the same fractal temporal range (or energy spectrum range) one can find, based on the resolution scale used in the simulation, a distinct evolution of the system, where, for *ω* values we see only one structure. With the increase of *ω*, we observe the formation of secondary structures along the orthogonal axis. It is worth noting that for intermediate values there is no interconnection between the original structure and the newly developed one (the *ω =* 6 case). This changes for larger values when the structures are well defined and there is a clear communication channel between them. Thus, in the fractal representation, with the change of the scale resolution, different scenes during the evolution of the system can be seen. The appearance of a secondary structure is a direct effect of the modulation phenomena seen in the 1-dimensional simulations presented in Figure 1. It is the interplay of the two-oscillation frequency, the real part which often defines the real, measurable space and the imaginary one, characterizing the interaction in the fractal space.

To test the validity of this new evolution scenario for complex system in a multifractal framework it is important to check the relevance of the mathematical approach. Over time, laser produced plasmas [19,20,21], and in general, low-temperature transient plasmas [22] have been promoted as great media in which non-linear and often chaotic behavior can occur. 

In Figure 2 we have represented a selection of ICCD images of a transient plasmas generated by ns laser ablation (10Hz, 532 nm, 10 J/cm) of a chromite target, placed in vacuum chamber (1Pa residual pressure). Extensive details on the dynamic of the chromite plasma can be seen in [23]. Briefly after irradiation, a high energetic plasma forms. The emission of the plasma is imaged by means of a cylindrical lenses optical system on the ICCD camera’s detector. An internal procedure is implemented to record images of the plasma at various moments in time. Supplementary investigations were performed on the plasma as was found to be characterized by 0.7 eV and two main structures along the main expansion axis expanding with 55 km/s and 14 km/s, respectively. Here the focus will be on short time evolution of the plasma (<500 ns) and the plume splitting phenomena. The generation of multiple plasma structure during laser produced plasma expansion has often been related to the existence of two types of ablation mechanism (thermal and electrostatic) which generate particle with different kinetic energy, thus inducing plasma structuring in the kinetic energy plane [24,25]. Recent work has also been dedicated to offering a hydrodynamic answer to this problem with promising results [26,27]. Each ICCD image from the sequence presented in Figure 3 has attributed two cross-sections along the main expansion axis (axial) and across the main expansion axis (transversal). During expansion the plasma increases its volume and the shape of the plasma considerable changes. This change is seen through the transversal cross-section where we see that the lateral shape of the plasma changed from quasi-Gaussian (below 250 ns) to a bi-lobbed one (above 250 ns). This change is attributed to a lateral plume splitting and confirms the simulated data from Figure 3, where we can see the generation of structure in the fractal space in a symmetrical manner across all axes. Although empirically, the lateral lobs are not always seen, they can act as a non-manifested potential ability in their evolution. The two lateral lobs appear as a result of the ablated particle scattering towards the edges of the multi-element plasmas, especially for expansion scenarios occurring the in atmospheres at relative high pressure as it is here (1 Pa). This result reflects well the simulated evolution in the multifractal medium where we see that the structuring of the fractal object does not have a particular direction when left unrestricted. The axial cross section reveals a classical image of the plasma with multiple peaks at 125 ns, which are better seen at longer evolution time (250 ns) where the separation based on their kinetic energy become clearer. Each maximum corresponds to a different energetic group of ions traveling with 55 and 14 km/s, respectively. The scattering and collisional processes are intimately related to the notion of fractality and fractal curve and the results here reaffirms the correlations between the fractal representation of the plasma and the empirical data.

## 6. Discussions

According with the above-presented results and some basic aspects of the Scale Relativity Theory presented in Appendix C, the SL(2R) group invariance of the motion equations naturally implies multifractalization through stochasticization (Markovian-type processes). In this context, the scale resolution can be correlated with the transitivity manifolds of the SL(2R) group.

From this perspective (the presence of Markovian-type processes), some correspondences can be found between our method and a standard one, based on the Wavelet Transform. This is mainly due to the fact that the most commonly employed Wavelet Function is the Gaussian (which can be induced by Markov-type processes) and its various order derivatives (details on the properties of the Wavelet Transform can be found in [28]).

## 7. Conclusions

Possible implications and consequences of using measurable Lie groups as invariance groups in the description of complex systems (at any scale resolution) are analyzed. From such perspective and based on Jaynes’ general observation according to which any circumstance left unspecified in the description of the dynamics of any complex system at any scale resolution has the concrete expression in the existence of an invariance group, in the present paper one specifies such unspecified circumstances, which result directly from the consideration of the canonical formalism. Then:
(i)Since the SL(2R) group invariances its elementary measure, it results that both the Hamiltonian and the Gaussian distribution density gain status of group invariant functions;(ii)The functionality of a maximum informational energy principle implies the fact that the Hamiltonian parameters gain statistical significance (for example, averages, variances, covariances, etc.);(iii)The class of statistical hypothesis that are specific to the Gaussians of the same average acts as group transitivity manifolds. In a broader context, the transitivity manifolds can be correlated with the multifractal-non-multifractal scale transitions. In particular, for the description of dynamics on monofractal manifolds through Peano type curves at Compton scale resolution, the constant which characterizes the transitivity manifolds can be put in correspondence with Planck constant;(iv)In the same transitivity conditions, joint invariant functions are generated through the isomorphism between the variables SL(2R) group and the parameters one;(v)For an ensemble of harmonic oscillators of same frequency, the unspecified circumstances unspecified circumstances amount to not knowing the amplitude and phase of each of the oscillators. This necessitates the recourse to a statistical ensemble by expliciting its element (the oscillator) at any scale resolution. This fact involves group parameters, the ensemble being traversed by the transformations of the Barbilian type group;(vi)Assimilating, at any scale resolution, the ensemble of oscillators with a multifractal type object, nonlinear behaviors of an ablation plasma are specified by simulation. The model thus designed has been validated based on certain experimental results. We note that such a theoretical model can be applied to analyze various dynamics of complex systems (for details, see [29,30,31]); and(vii)The novelty of our method, when compared to other models, such as the ones from [32,33] resides in the fact that fractalization through stochasticization is imposed through group invariance, situation in which the group’s transitivity manifolds of SL(2R)-type can be correlated with the scale resolution.

## Figures and Tables

**Figure 1 entropy-24-00484-f001:**
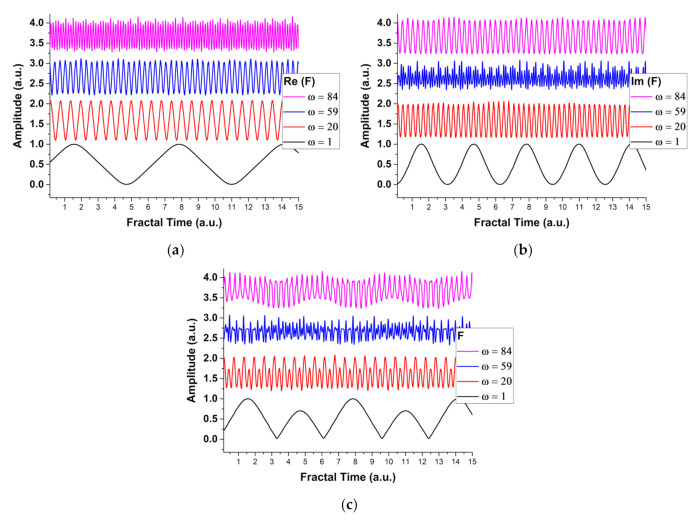
Temporal sequences at various interaction scales (1, 20, 59, 84) for (**a**) Re (*F*), (**b**) Im (*F*), and (**c**) *F*.

**Figure 2 entropy-24-00484-f002:**
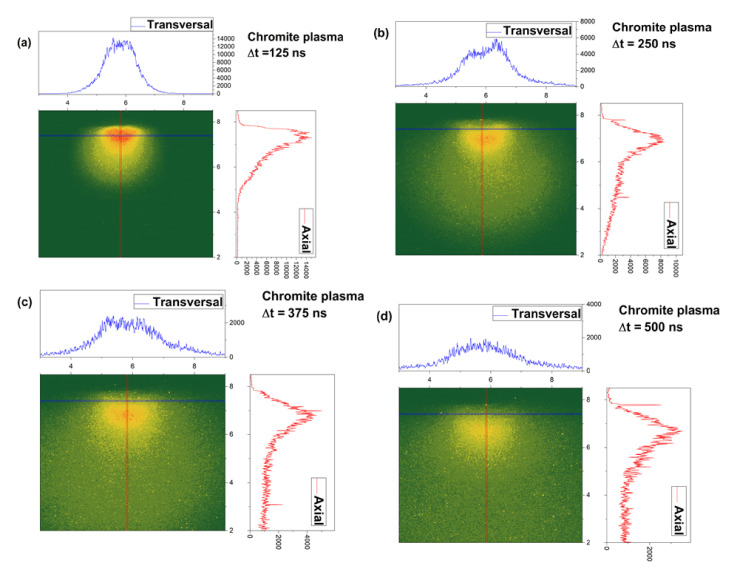
ICCD Images of a laser produced Chromite plasma with the corresponding cross-sections on axial and transversal directions.

**Figure 3 entropy-24-00484-f003:**
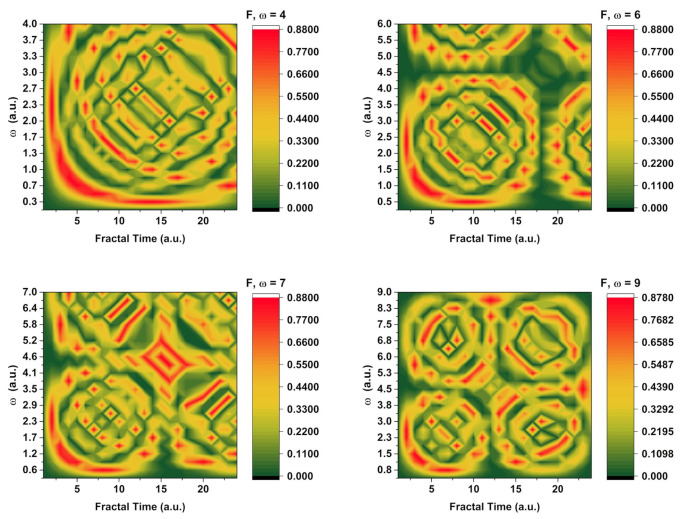
2D contour plot of typical evolution sequence for complex system dynamics in a fractal representation.

## Data Availability

Not applicable.

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
