# Peer review of "Implications and Consequences of SL(2R) as Invariance Group in the Description of Complex Systems Dynamics from a Multifractal Perspective of Motion"

_entropy, 2022, doi:10.3390/e24040484_

Round 1

Reviewer 2 Report

The main problem with this manuscript is that the authors do introduce a feature extraction technique - but do not manage to properly describe its properties. The authors pay a lot of attention to the multi-fractal interpretation of the introduced visualisation technique. However, the multi-fractality is exactly the weakest point of this manuscript. 

Usually, the introduction of any algorithm capable to reveal multi-fractal features of a dynamical system should follow some general rules. Now, the authors just introduce some simple computational results without any deeper analysis on the properties of the introduced algorithm. Fig. 2 does not help to make the situation better. Changing the scale of the vertical axis (and claiming the fractality of the horizontal axis) just helps to vary the size of the observation window. 

Similar results (in terms of the "fractality") could be reproduced by 2D FFT, or 2D wavelet transforms. 

All claims of the authors in that sense are not convincing. 

The second problem of this manuscript is related to the fact that the manuscript lacks comparisons with other alternative techniques (feature extractions from the footprints of dynamical systems).

The third problem is that the topic of this study is not related to entropy or its analytical or computational variations. 

Round 2

Reviewer 1 Report

The paper has improved and the authors managed to address my concerns.
In my view, the paper can be accept.

Reviewer 2 Report

The authors did perform a revision of the original manuscript according to the recommendations from the reviewers. The appendixes and the clarification of the novelty of the contribution did help to improve the manuscript. However, some questions had not been answered, especially in respect to the comparison with alternative multi-fractal algorithms. The authors should elaborate more on these questions. It is not the best way to face such a request by guiding the reviewer to read papers previously published by the same co-authors. 

Another question which is arising after a careful reading of the manuscript also requires a special attention. Lie groups, the SL(2R) invariance, complex multi-fractal dynamics are specialised topics in pure mathematics. However, the affiliations of authors are really strange. Department of Physical and Occupational Therapy, Faculty of Dental Medicine - that makes the eyebrows to raise. The authors should carefully explain the contributions of each individual author in order to eliminate concerns regarding the possibility that the list of co-authors does contain "gift" co-authors. 

Round 3

Reviewer 2 Report

The revised manuscript can be now recommended for publication.